# NGF Prevents Loss of TrkA/VEGFR2 Cells, and VEGF Isoform Dysregulation in the Retina of Adult Diabetic Rats

**DOI:** 10.3390/cells11203246

**Published:** 2022-10-15

**Authors:** Elena Fico, Pamela Rosso, Viviana Triaca, Marco Segatto, Alessandro Lambiase, Paola Tirassa

**Affiliations:** 1Institute of Biochemistry and Cell Biology, National Research Council (CNR), Department of Sense Organs, Sapienza University of Rome, Viale del Policlinico 155, 00161 Rome, Italy; 2Institute of Biochemistry and Cell Biology, National Research Council (CNR), International Campus A. Buzzati Traverso, Via E. Ramarini 32, Monterotondo, 00015 Rome, Italy; 3Department of Biosciences and Territory, University of Molise, Contrada Fonte Lappone, 86090 Pesche, Italy; 4Department of Sense Organs, Sapienza University of Rome, Viale del Policlinico 155, 00161 Rome, Italy

**Keywords:** NGF, VEGF, TrkA, eye drops, streptozotocin, diabetic retinopathy

## Abstract

Among the factors involved in diabetic retinopathy (DR), nerve growth factor (NGF) and vascular endothelial growth factor A (VEGFA) have been shown to affect both neuronal survival and vascular function, suggesting that their crosstalk might influence DR outcomes. To address this question, the administration of eye drops containing NGF (ed-NGF) to adult Sprague Dawley rats receiving streptozotocin (STZ) intraperitoneal injection was used as an experimental paradigm to investigate NGF modulation of VEGFA and its receptor VEGFR2 expression. We show that ed-NGF treatment prevents the histological and vascular alterations in STZ retina, VEGFR2 expression decreased in GCL and INL, and preserved the co-expression of VEGFR2 and NGF-tropomyosin-related kinase A (TrkA) receptor in retinal ganglion cells (RGCs). The WB analysis confirmed the NGF effect on VEGFR2 expression and activation, and showed a recovery of VEGF isoform dysregulation by suppressing STZ-induced VEGFA_121_ expression. Reduction in inflammatory and pro-apoptotic intracellular signals were also found in STZ+NGF retina. These findings suggest that ed-NGF administration might favor neuroretina protection, and in turn counteract the vascular impairment by regulating VEGFR2 and/or VEGFA isoform expression during the early stages of the disease. The possibility that an increase in the NGF availability might contribute to the switch from the proangiogenic/apoptotic to the neuroprotective action of VEGF is discussed.

## 1. Introduction

Among the retina trophic factors, the neurotrophin nerve growth factor (NGF) [1,2] and vascular endothelial growth factor A (VEGFA), the first discovered VEGF family member [3,4], have been demonstrated to be pivotally important in supporting functional retinal integrity [5,6]. Altered VEGF/NGF synthesis and/or changes in the expression and activation of their specific receptors; namely, vascular endothelial growth factor 2 (VEGFR2) and tropomyosin-related kinase A (TrkA), respectively, have also been found to contribute to retina degenerative events, involving both the vascular system and the neurons similarly occurring in diabetic retinopathy (DR) [7,8].

In the retina, VEGF is mainly described as a proangiogenic factor, and the synthesis dysregulation of the A form of VEGF in the retinal pigmented epithelium (RPE) upon the intracellular cascade, initiated by Rho-associated coiled coil kinase 1 (Rock1) activation, is one of the trigger events in DR vascular impairment [9].

Five proteins are produced by VEGFA splicing: VEGFA_121_ and VEGFA_165_, which are secreted proteins, and VEGFA_145_, VEGFA_189_ and VEGFA_206_ that are strongly cell-associated [10]. VEGFA’s effects on the retina depend on the relative amounts of VEGFA_121_ and VEGFA_165_, and their interaction with vascular endothelial growth factor receptor 2 (VEGFR2) [11,12,13,14], thus underlining the importance of VEGF regulation in pathological conditions.

Beside the RPE, VEGFA also acts in an autocrine and paracrine manner on retinal neurons, including retinal ganglion cells (RGCs) [15], which also express VEGFR2 [16]. The evidence that VEGFA might exert a protective effect on RGCs, while VEGFA antagonism reduces their axonal transport [17] and activates cellular death pathways [6], supports the idea that changes of VEGFA in neuroretina might favor neurodegenerative events in DR [18].

The mechanism by which VEGFA might exert neuroprotective effect remains to be clarified, but the findings that VEGFA antagonism downregulates neurotrophin receptors, modulates the proform of NGF (proNGF), and stimulates apoptotic signals in the retina [6], suggests that the interaction between NGF and VEGF participates in the retinal maintenance during health and disease.

NGF neuroprotection in the retina, as well as in the brain, is mediated by the interaction with its selective receptor TrkA, and its downstream survival pathway, including serine/threonine kinase named Akt as well as the extracellular-signal-regulated kinase (ERK) [19,20,21,22].

Reduction or lack of available mature NGF results in RGC loss, which is manifested by a decrease in TrkA expression/activation, and Akt and/or ERK signaling disruption, as observed in different models of retina degeneration [21,22,23]. In contrast, ocular administration of exogenous NGF reverses TrkA impairment and counteracts gliosis, pro-apoptotic proNGF increase, and consequent RGC loss, following optic nerve lesions, and diabetes [1,19,24].

Although NGF might exert its effects on endothelial cells to induce angiogenesis [25], no microvessel alterations have been reported in the eyes following ocular NGF administration in humans [26,27]. Moreover, the recovery of brain vascular alterations has been found in animal models, including diabetes [24], suggesting that the crosstalk between NGF and VEGFA might participate in retinal cell survival.

To explore this hypothesis, the administration of eye drops containing NGF (ed-NGF) to adult Sprague Dawley rats receiving intraperitoneal injections of streptozotocin (STZ) was used as an experimental paradigm to investigate the role of the NGF/TrkA system in VEGFA and VEGFR2 metabolism. STZ rats represent a widely used and extensively characterized DR animal model [28]. In this model, degeneration of the neuroretina is already visible at 2–3 weeks after diabetes induction, while significant vascular alterations, and VEGFA upregulation occur at 6–8 weeks post-injection [4]. Thus, to investigate whether the increased availability of mature NGF interferes with VEGFA-related events in the retina, ocular NGF treatment was administered at the DR onset and repeated for two weeks twice a day.

The data reported by this study confirm that topically administered ed-NGF is able to counteract the STZ-induced retina pathology. Further, based on ed-NGF driven modulation of VEGFA isoform expression and VEGFR2 distribution and activation, our data also pinpoint a cooperative NGF/VEGF activity in sustaining retinal cell function and RGC survival at the early stages of diabetic pathology.

## 2. Materials and Methods

### 2.1. Animal Model of Diabetes (STZ) and Topical NGF (Eye Drop NGF) Treatment

The animals used in this study were 30 adult male, pathogen-free, Sprague Dawley (SD) rats (200–250 g, Harlan-Nossan, Italy). Since neurotrophins and their receptors can vary across the estrous cycle in both the central nervous system and peripheral tissues [29,30], and that adult female rats have to be in the same phases of both cycles in order to avoid overlapping effects due to changes in hormone levels, only adult male rats were included in this experimental setting.

A single intraperitoneal (IP) injection of streptozotocin (STZ, Sigma, St Louis, MO, USA; 60 mg/kg body weight dissolved in physiological solution) was used to induced diabetes in rat. The animals were allowed to drink a 10% dextrose solution overnight and then placed following the standard laboratory conditions. To evaluate glucose level in blood at certain time points such as day 0, which indicated prior to STZ IP injection or saline solution, and after STZ IP injection, a glucometer (Contour XT, Bayer, Germany) was used. Diabetes was diagnosed for rats whose blood glucose levels rose above 250 mg/dL, and thus these animals were included in the experimental design for the evaluation of ocular NGF administration effects. The mortality rate was ~20%. For ocular treatment, in a physiological solution (0.9% sodium chloride), purified NGF obtained from male mouse submaxillary salivary glands [31] was dissolved to a concentration of 200 µg/mL.

Rats were divided into three experimental groups: the control (CTRL) group included healthy rats receiving IP injections with saline solution, the STZ group was composed of diabetic rats receiving two drops of physiological solution per eye, and the STZ+NGF group included STZ rats, which received two drops (10 µL each) of 200 µg/mL NGF to both eyes.

As previously found, the effects of STZ on the retina of SD adult rats, in terms of both loss of vessel integrity and loss of RGCs, as well as VEGF increase, are progressive and reach significant levels between 4 and 8 weeks post-injection [4]. Thus, to evaluate the protective effects of ed-NGF, the ocular treatment started 6 weeks after diabetes induction, and was repeated twice a day for 2 weeks. Sacrifice was performed by decapitation, under previa anesthesia at 8 weeks after diabetic induction. 

For biochemical analysis, the eyes were removed, and the retina was isolated on ice under a microscope. The retinal samples were stored at −80 °C until use. The eyes from each rat were randomly assigned, with one to the biochemical and one to the histological evaluation. For histology and immunofluorescence (IF) studies, the eyes were fixed in 4% paraformaldehyde (PFA) in phosphate-buffered saline (PBS) solution for 24 h and then dehydrated in 20–30% sucrose solution. The eyes/retinas were then processed as described below.

Rats were maintained in a 12 h light–dark cycle and provided with food and water ad libitum during the whole experimental period. The guidelines indicated were approved by the intramural Committee and Institutional Guidelines of the Italian National Research Council in conformity with National and International laws (EEC Council Directive 86/609, OJ L 358, 1, 12 December 1987) and were followed for the housing, care, and experimental procedures, and all efforts were taken to limit the number of rats. Moreover, the ocular procedures were in accordance with the ARVO statement for the use of animals in ophthalmic and vision research.

### 2.2. Lysate Preparation from Rat Retinal Samples

To perform protein extraction, homogenization of tissue samples by ultrasonication in RIPA buffer (50 mM Tris–HCl, pH 7.4; 150 mM NaCl; 5 mM EDTA; 1% Triton X-100; 0.1% SDS; 0.5% sodium deoxycholate; 1 mM PMSF; 1 mg/mL leupeptin) was used. Samples were then kept in a cold room on a rotary shaker for 2 h to allow complete tissue disaggregation and cell lysis. After, the tissue lysates were centrifuged at 13,000 rpm for 30 min at 4 °C to remove tissue debris. The supernatants collected were then used to measure the total protein concentration by performing a BioRad Protein Assay (500-0116, Bio-Rad Laboratories, Richmond, CA, USA). All lysate samples were boiled for 5 min before loading on to subsequent SDS-PAGE and Western Blots (WBs).

### 2.3. Western Blot

SDS-PAGE and transfers to nitrocellulose membranes were used to separate and analyze samples (20–50 µg of total protein). Samples were dissolved in loading buffer (0.1 M Tris-HCl buffer, pH 6.8, containing 0.2 M dithiothreitol, 4% sodium dodecylsulfate, 20% glycerol, and 0.1% bromophenol blue). The membranes were incubated for 1 h at room temperature (RT) with 5% non-fat dry milk dissolved in 10 mM Tris (pH 7.5), 100 mM NaCl, and 0.1% Tween-20 (TBS-T), washed three times for 10 min each in TBS-T solution, and then incubated overnight at 4 °C with primary antibodies against: Rock1 (1:1000, #4035; Cell Signaling Technology), VEGFA (1:1000, C-1: sc-7269), VEGFR2 (1:1000, Flk-1 D-8: sc-393163), pVEGFR2 (1:1000, #MABS191), TrkA (1:1000, sc-118), pTrkA (1:500, #9141), proNGF (1:500, ANT-005), Akt (1:1000, #2920), pAkt (1:1000, #9271), ERK1/2 (1:1000, sc-514302), pERK1/2 (1:1000, sc-7383). The secondary antibodies used were horseradish peroxidase (HRP)-conjugated anti-rabbit or anti-mouse IgG (Bio-Rad Laboratories), and then the membranes were developed using enhanced chemiluminescence (ECL) detection (Bio-Rad Laboratories) and exposure to Amersham Hyperfilm (GE Healthcare) or image acquisition through an iBright Western Blot Imaging System (FL1500, Thermofisher Scientific, Singapore).

The Western Blot analysis was performed in triplicate of each experimental condition, and evaluation of the marker expression was performed by using ImageJ (1.52t, National Institute of Health, Bethesda, MD) software for Windows. β-Actin-HRP (1:10,000, sc-47778) was used as a housekeeping antibody for protein loading normalization. Values were obtained from protein band/related β-Actin band ratio expressed as arbitrary units (a.u.) and mean ± SD. 

### 2.4. Confocal Imaging Analysis

#### 2.4.1. Histology and Immunofluorescence

For toluidine blue 0.1% and immunofluorescence (IF) staining, the eyes were first embedded in Tissue-Tek OCT compound, sectioned at 25–30 μm thickness with a cryostat (Leica Microsystems GmbH, Nussloch, Germany), and mounted on superglass slides. IF was performed on eye sections, containing the retina layers, rinsed with PBS 0.1M and then incubated with primary antibodies against VEGFR2 (1:100, Flk-1 D-8:sc-393163) and TrkA (1:100, sc-118), or VEGFR2 (1:100, Flk-1 D-8:sc-393163) and NeuN (1:250, #12943, Cell Sig-naling Technology), or Isolectin Griffonia simplicifolia biotin-conjugate (GS-IB_4_, Invitrogen Corporation, USA) to identify blood vessels on retina sections (*n* = 3 per group), overnight at 4°C. The primary antibodies were diluted in PBS 0.1M with 0.1% Triton X-100 (PBST). The primary antibodies were diluted in 0.1 M PBS with 0.1% Triton X-100 (PBST). The following day, sections were washed three times for 10 min each in 0.1 M PBS, and then incubated for 2 h at RT with Alexa Fluor anti-rabbit 555 (1:500, A31572, Invitrogen Corporation, Carlsbad, CA, USA) fluorochrome-conjugated secondary antibodies or Alexa Fluor anti-mouse 488 (1:500, A21202, Invitrogen Corporation, USA) diluted in 0.1 M PBS. Streptavidin-AV-Alexa 594 (1:100; Invitrogen Corporation, USA) was diluted in 0.1 M PBS and used for GS-IB4 detection. Sections were then washed three times for 10 min each in 0.1 M PBS and coverslips were mounted on double IF glass slides (Superfrost Plus; Thermofisher) with the antifading VECTASHIELD mounting medium with 4′,6-diamidino-2-phenylindole (DAPI) (Vector Laboratories, H-1200), while for GS-IB4 glass slides (Superfrost Plus; Thermofisher), the coverslips were mounted by using the antifading VECTASHIELD mounting medium (H-1000). All steps were performed under gentle shaking. All sections were analyzed and VEGFR2/TrkA and VEGFR2/NeuN representative images were captured using the laser scanning confocal microscope Olympus FV1200, visualized with the FV10-ASW software (Version 4.2, Olympus, Tokyo, Japan), composed with Adobe illustrator and finalized in Photoshop.

#### 2.4.2. VEGFR2 Quantitative Analysis

The quantification of VEGFR2 immunofluorescence was analyzed in the retinal cryosections immunolabelled for VEGFR2 (green)/NeuN(red) by confocal imaging using the oil 40× objective at a resolution of 1024 × 1024 pixels. Briefly, upon background normalization and thresholding in the green channel, a binary mask was created by using Fiji (ImageJ, NIH, Baltimore). The integrated density and the percentage area of VEGFR2 immunofluorescence were measured in approximately six randomly selected, non-overlapping areas (200 × 200 μm) spanning all the retina layers (from ONL to GCL) and averaged per retinal cryosection. At least three retinal cryosections per experimental group were used. Data were reported in the graphs as the percentage of CTR and reported as the mean + standard error of mean (SEM).

### 2.5. Retina Vasculature Analysis

Imaging analysis of GS-IB4 IF slides was performed with the Nikon NIS-Elements AR 2.30 program. For the vasculature assessment, the method of area fraction previously described [24] was used, counting the area occupied by blood vessels with respect to the total measured area in 10× magnification images. Further, the GS-IB4 immunoreactive vessels (identified objects) having areas of <10; between 10–50; 50–100 and >100 µm^2^ were counted in each section/group. The values are expressed as the mean percentage vessel area ± S.D., and as vessel numbers/mm^2^ ± standard deviation (S.D.).

### 2.6. Statistical Analysis

One-way analysis of variance followed by Tukey’s post-hoc test was used to analyze the data, values of *p* < 0.05 were considered to indicate a significant difference. Statistical analysis was performed using GraphPad InStat3 (GraphPad, La Jolla, CA, USA) and by Statview (version 5.0, SAS Institute Inc, Cary, NC, USA) for Windows.

## 3. Results

### 3.1. Histological Alterations in the Retina of STZ and STZ+NGF

Figure 1A,B report the diabetic effects on retinal histology and vasculature with or without the administration of ed-NGF. Comparing the STZ group with STZ+NGF or healthy (CTRL) rats, toluidine blue stain revealed retinal changes (Figure 1A). Tissue damage was indeed observed in all the retina layers of STZ rats, with a cellular loss in the ganglion cell layer (GCL) and the inner nuclear layer (INL) (middle picture, Figure 1A). A better identification of cellular bodies in the GCL is observable following NGF administration (STZ+NGF) in right picture, Figure 1A.

The retinal vasculature was identified by using isolectin GS-IB_4_ staining. As is shown in Figure 1B, a fragmentation of the retina vessels of the superficial and intermediate vascular plexus (SVP and IVP, respectively), was associated with an increase in staining in the RPE/choroid area, and was observable in diabetic rat retinas (middle picture, Figure 1B), when compared to both CTRL and STZ+NGF groups (left and right pictures, respectively, Figure 1B).

The computer analysis confirmed the microscope observations showing significant reductions of the vessel area fraction, and an increased number of small vessels (area <50 µm^2^) in STZ retinas with respect to both CTRL and STZ+NGF groups (Figure 1C,D, respectively).

### 3.2. Vascular Marker Expression in Retina

The vascular impairment in STZ retina was also evaluated by analyzing the expression of VEGFA and Rock1, which are both involved in retinal microvascular damage and angiogenesis [32] (Figure 2A–F). In accordance with previous studies [12], we found that different VEGFA isoforms, both in monomeric and dimeric status, can be detected in rat retina by Western Blot (WB) analysis. The dimeric VEGFA_121_, VEGFA_165_ and VEGFA_189_, corresponding to bands at 30 kDa, 46 kDa, and 50 kDa, respectively, were the most expressed forms detectable in our experimental conditions (Figure 2A–D). As is shown by the representative membrane, an increase of the band at 30 kDa (VEGFA_121_) was recognizable in STZ retina, while it was barely expressed in the CTRL and STZ+NGF groups (Figure 2A). The statistical comparison of the densitometric analysis confirmed this observation (*p* < 0.05, Figure 2B), and showed that VEGFA_121_ (Figure 2B) and VEGFA_165_ (Figure 2C) were significantly decreased (*p* < 0.05) and increased (*p* < 0.05), respectively, after ed-NGF (STZ+NGF) treatment compared to diabetic conditions (STZ). No differences between the groups were found when analyzing VEGFA_189_ expression levels (Figure 2D).

Figure 2E,F report how retinal Rock1 levels are affected in diabetic conditions and after NGF ocular administration. A very high expression Rock1 level, almost a 3-fold increase compared to the CTRL value, was found in STZ retina (*p* < 0.001), while a significant decrease was observed in STZ+NGF (*p* < 0.001) compared to the diabetic group (Figure 2F).

### 3.3. Expression Levels of Total and Phosphorylated VEGFR2

The WB analysis of VEGFR2 expression and phosphorylation status (pVEGFR2) showed that VEGFR2 decreased in diabetic rats (STZ) compared to CTRL, and significantly increased after NGF ocular treatment (STZ+NGF) when compared to the STZ group (*p* < 0.001; Figure 3A,B). pVEGFR2 level was found to be significantly decreased in STZ, and significantly increased after NGF administration (STZ+NGF) compared to CTRL (*p* < 0.01 and *p* < 0.05, respectively; Figure 3C). Further, in the STZ+NGF group, pVEGFR2 levels increased significantly when compared to STZ (*p* < 0.001; Figure 3C).

No significant differences were found when comparing the ratio of VEGFR2 phosphorylation and the related total form (pVEGFR2/VEGFR2) of diabetic rat retina *versus* STZ+NGF group, while the same ratio of both groups decreased when compared to CTRL (Figure 3D).

### 3.4. Effects of STZ and NGF Eye Drops on TrkA Expression and Activation

The expression and activation of TrkA levels in STZ and STZ+NGF rat retina is shown in the representative gels of Figure 4A. No significant changes in the expression of total (Figure 4B) and phosphorylated TrkA (Figure 4C), as well as in the pTrkA/TrkA ratio (Figure 4D), were found in STZ and STZ+NGF retina when compared to CTRL.

### 3.5. NeuN, VEGFR2 and TrkA distribution in rat retina

Double immunofluorescence was performed to study the distribution of VEGFR2 (green) and neuronal nuclei (NeuN, red, Figure 5), as well as VEGFR2 (green) and TrkA (red, Figure 6) in rat retinal layers belonging to the three experimental groups: CTRL, diabetic (STZ) and STZ+NGF. The panels in Figure 5A–C shows the mature neuron immunoreactivity for NeuN together with the VEGFR2 labelling through the main layers: n layer (ONL), inner nuclear layer (INL), and the ganglion cell layer (GCL). In the diabetic rat group (Figure 5B), reduced VEGFR2 expression in GCL was detected as well as its recovery after NGF eye drops treatment (STZ+NGF, Figure 5C), almost comparable to that in the healthy retina (CTRL). The islets A’–C’ (Figure 5) report the high magnification of the VEGFR2/NeuN staining to better appreciate the detail of VEGFR2 expression decrease in STZ neurons compared to CTRL and to the STZ+NGF rat retina groups. The graphs of VEGFR2 integrated density and the VEGFR2 percentage area shown in Figure 5D and E, respectively, confirm the VEGFR2 expression trend observed in confocal microscopy images, demonstrating the significantly decrease in the STZ when compared to CTRL (*p* < 0.05), and recovery after ed-NGF administration to diabetic retinas (*p* < 0.01).

Figure 6 shows the reduction of VEGFR2 immunoreactivity, particularly in the GCL and in INL of the STZ rat group, as well as TrkA labelling (Figure 6B) even if the NGF-related receptor remains visible in RGC layer (GCL). Conversely of both CTRL and STZ+NGF groups, small cells expressing VEGFR2 were observed in the INL of the STZ retina. In Figure 6C, following NGF eye drop administration (STZ+NGF), the organization of the retinal layers appears similar to the CTRL group (Figure 6A), and an increase of VEGFR2 immunoreactivity compared to the STZ group was observed in the GCL. The islets A’-C’ show at a higher magnification the STZ-induced abnormalities in the organization of the retinal layers together with the reduced expression of VEGFR2 and, to a lesser extent, TrkA, showing the major effects in the GCL (yellow rectangle) where the co-localization of TrkA and VEGFR2 was almost undetectable. In contrast, upon ed-NGF treatment, TrkA and VEGFR2 co-expression was high in STZ rat retina upon ed-NGF treatment.

### 3.6. Retinal Expression of ProNGF Levels

The WB analysis of proNGF was performed by using an antibody able to recognize different forms of the proneurotrophin depending on distinct stages of proform maturation and on its glycosylation status (Figure 7A–C). A significantly increased level of the 26 kDa proNGF (*p* < 0.01), and its glycosylated form (*p* < 0.001) at 40 kDa, was observed in STZ retina when compared to CTRL (Figure 7B,C, respectively). This effect was counteracted by ed-NGF treatment, as reduced level of both 26 kDa and 40 kDa proNGF were recorded in STZ+NGF animals compared to the STZ group (*p* < 0.05 and *p* < 0.01, Figure 7B,C, respectively).

### 3.7. Intracellular Signal Activation in STZ Retina

The intracellular pathways activated by VEGFA and NGF/TrkA were investigated by analyzing the expression levels and phosphorylation statuses of ERK1/2 and Akt in STZ and STZ+NGF rats (Figure 8A–H). A significant increase in phosphorylated ERK (pERK) levels was found when comparing STZ+NGF and CTRL groups (*p* < 0.01), while a slight increase was observed in STZ retina compared to healthy rats (Figure 8C). Total ERK (ERK) did not show any significant alterations across all the experimental groups (Figure 8B). In Figure 8D, upregulation of the pERK/ERK ratio was observed in diabetic rats treated with ocular NGF (STZ+NGF), resulting on a two and three times increase when compared to both STZ and CTRL (*p* < 0.01).

In contrast to ERK signaling, total Akt and phosphorylated Akt (pAkt) levels (Figure 8E–G) were reduced following STZ induction when compared to CTRL (*p* < 0.001 and *p* < 0.01, Figure 8F,G, respectively). Upon administration of ed-NGF (STZ+NGF), Akt total expression, phosphorylation status (pAkt), and pAkt/Akt ratio were recovered when compared to diabetic rat retina (*p* < 0.001, *p* < 0.001 and *p* < 0.05, Figure 8F–H, respectively), and rescued comparable to control levels (STZ+NGF *versus* CTRL).

## 4. Discussion

The main goal of the present study was to further investigate the role of VEGF/NGF interaction in the neuroretina by analyzing the effects of ed-NGF treatment in the STZ model of DR. As already showed by means of different models of retinal degenerations, ed-NGF are able to counteract RGC loss, and exerts protective and reparative actions through the normalization of neurotrophins and expression of TrkA receptors and its related intracellular pathways [21,23,33,34].

To confirm, we now show that ed-NGF treatment, during the first few weeks after STZ-injection, results in the attenuation of histological and vascular alterations associated with a rebalance of VEGFA and VEGFR2 expression, and s reduction in inflammatory/pro-apoptotic signals. In addition, to the best of our knowledge, the present study is one of the first to demonstrate that ed-NGF might act by regulating the STZ-induced alterations of VEGFA isoform expression in the retina, potentially contributing to RGC loss in the early diabetic stages.

Within two months after STZ injection, the major histological alterations in the retina are detectable in the neuronal components, the INL and the GCL, where damage and death of bipolar, amacrine cells, RGCs, and Müller cells have been reported [35,36,37].

Our confocal observations that VEGFR2 is decreased in STZ neuroretina, particularly in the GCL, confirmed by NeuN/VEGFR2 labelled RGCs and the related VEGFR2 integrated density and VEGFR2 percentage area analyses (Figure 5), support previous evidence that this cell population is highly vulnerable to diabetes in the early stages of the pathology [38,39].

In line with this, we also found that the STZ-induced alterations of the VEGFR2 was associated with a decrease in trophic and survival intracellular signals (Akt and ERK), and increased proNGF expression. Ed-NGF treatment in diabetic rats reversed the downregulation of intracellular signals, and inhibited the accumulation of proNGF, thus confirming that the increase in NGF availability counteracts the events which might favor the progression of retinal degeneration, as suggested by previous works [1,24]. 

The findings that two weeks of ocular NGF administration also prevented the reduction of TrkA and VEGFR2 activation induced by STZ condition, and the loss of TrkA/VEGFR2 co-expression in RGCs, also sustain the idea that the RGC population maintains the ability to respond to NGF treatment when performed during the early disease stages. 

Recently, it was observed that the neurodegeneration phase, involving RGCs in particular, occurs firstly in DR outcome, precedes the vascular stage, and potentially could be prevented using NGF as a neuroprotective strategy [40], further supports our study. Indeed, in association with the effects on neuroretina, we also found that ed-NGF counteracted vessel fragmentation and rebalanced the expression of VEGFA isoforms in STZ retina.

Our study shows that VEGFA isoforms are differentially expressed in the diabetic retina as reported in Figure 2A–F. In fact, we observed that VEGFA_121_ isoform is highly expressed in diabetic, but it is almost undetectable in both CTRL and STZ+NGF retina, and that ed-NGF induces an increase of the VEGFA_165_ isoform (Figure 2C), indicating that NGF might not only produce changes in total VEGFA [36], but also regulates the expression of the three VEGFA isoforms detected: VEGFA_121_, VEGFA_165_ and VEGFA_189_ (Figure 2A–D). Although this issue needs to be further investigated using selective anti-VEGFA antibodies to discriminate among the different isoforms, these first data might pave the way to a novel approach toward the NGF/VEGFA crosstalk in retinal physiology and disease.

Indeed, as largely demonstrated, the VEGFA isoforms exert diverse and/or divergent effects on their targets, including endothelial cells, glia and neurons, resulting in promotion or inhibition of vascularization, survival and trophism [41]. VEGFA_121_ and VEGFA_165_ also show a diverse diffusibility and affinity to VEGFA receptors [13], and their target cell outcomes highly depend on the presence and phosphorylation status of VEGFR2 [42].

Furthermore, the existence of VEGFA isoform variant, VEGF_xxxb_, which can have anti-apoptotic and anti-vascular (protective) functions, suggests the possibility that, directly or indirectly, ed-NGF might affect VEGFA splicing, and therefore contribute in regulating the fate of VEGFA target cells in both physiological and pathological conditions.

In this context, it is relevant that we found ed-NGF treatment also resulted in a decrease in microvessel fragmentation in STZ (Figure 1B–D), and reduced expression of Rock1 (Figure 2E,F), a marker of endothelial cell dysfunction [9], as well as Müller cells in diabetic retina [43].

The double reparative effect of ed-NGF on vascular and neuronal compartments was already observed in the brain of adult STZ rats with encephalopathy [24], and by Moser et al. in a model of cholinergic degeneration [44], indicating that NGF, beside its role on neurons, might act by contrasting glia and endothelial cell dysfunction, as well as inflammation, sustaining the neurodegenerative process.

Since VEGFR2 is expressed by Müller cells, but also in microglia and endothelial cells, the possibility that ed-NGF might exert its effect through cells expressing VEGFA isoforms/VEGFR2 is reasonable and needs more extensive investigation.

In conclusion, our study shows that increased availability of mature NGF is able to revert STZ-induced alterations of VEGFA and VEGFR2 expression in adult retina and activates TrkA-mediated anti-apoptotic and neuroprotective pathways which might prevent the loss of RGCs and the accumulation of microvascular defects.

However, it is important to take into consideration that in our study the treatment with ed-NGF started in the early stages of retinal degeneration following STZ administration, when the gliosis is still not present, and there is a low grade of vascular impairment. Since NGF has also been shown to stimulate angiogenesis, and activate glial cells in a dose-related manner [45], the possibility that ed-NGF has detrimental effects on diabetic retina when administrated at high doses, and/or in the late phase of the disease, cannot be excluded and deserves further study.

## Figures and Tables

**Figure 1 cells-11-03246-f001:**
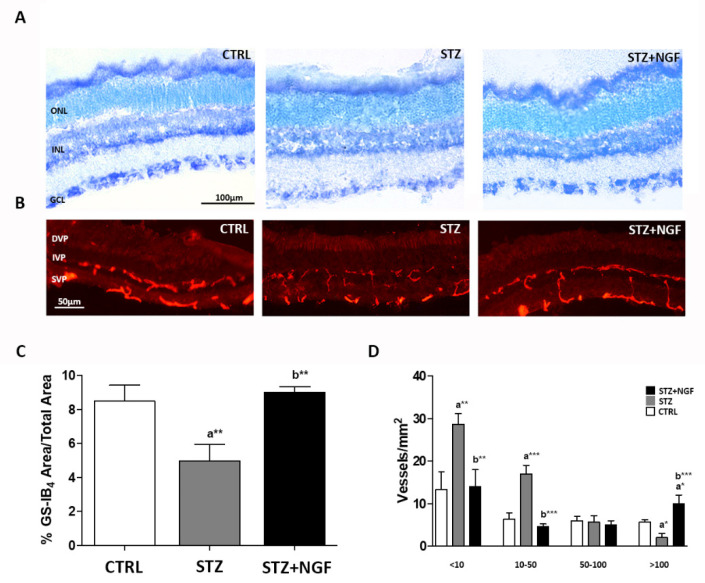
(**A**–**D**). Retinal histology and vasculature are affected by diabetes (STZ) and NGF eye drops administration (STZ+NGF). (**A**) Toluidine blue stain (black bar = 100 µm) revealed changes when comparing inner nuclear layer (INL) and ganglion cell layer (GCL) of STZ rat retinas with the healthy (CTRL) and STZ+NGF groups, while minor differences were observed between CTRL and STZ+NGF. (**B**) The distribution of GS-IB_4_-positive vessels in the retina (red stain), white bar = 50 µm. A fragmentation of retinal vessels in the area between the superficial vascular plexus (SVP) and intermediate vascular plexus (IVP) was observable in the diabetic retina with respect to healthy and STZ+NGF groups, respectively. (**C**,**D**) The results of computer analysis of vasculature reported the effects of STZ on the area fraction (**C**) and on the size distribution of vessels/mm^2^ (**D**) expressed as mean ±SD. Statistical significance is indicated with * *p* < 0.05, ** *p* < 0.01, *** *p* < 0.001. “a” vs. CTRL, “b” vs. STZ. ONL, outer nuclear layer; DVP, deep vascular plexus.

**Figure 2 cells-11-03246-f002:**
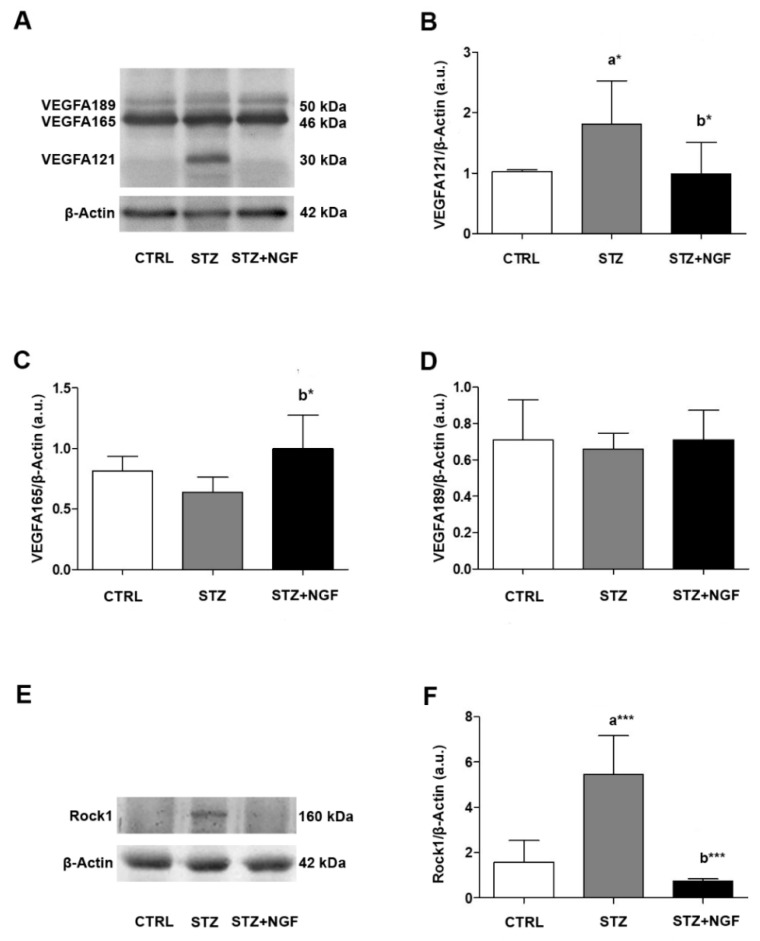
(**A**–**F**). VEGFA isoforms and Rock1 in diabetic rat retina before and after NGF ocular treatment. Levels of VEGFA isoforms 121, 165, 189 (30 kDa, 46 kDa and 50 kDa, respectively) and Rock1 were affected by diabetes (STZ) and NGF eye drops administration (STZ+NGF) as shown by Western Blots (**A**,**E**) and densitometric analyses (**B**–**D**,**F**). β-Actin expression was used to normalize sample variability; a.u. indicates arbitrary units. The results are expressed as mean ±SD. Statistical significance is indicated with * *p* < 0.05, *** *p* < 0.001. “a” vs. CTRL, “b” vs. STZ. CTRL, healthy rats; STZ, streptozotocin; STZ+NGF, streptozotocin+nerve growth factor eye drops.

**Figure 3 cells-11-03246-f003:**
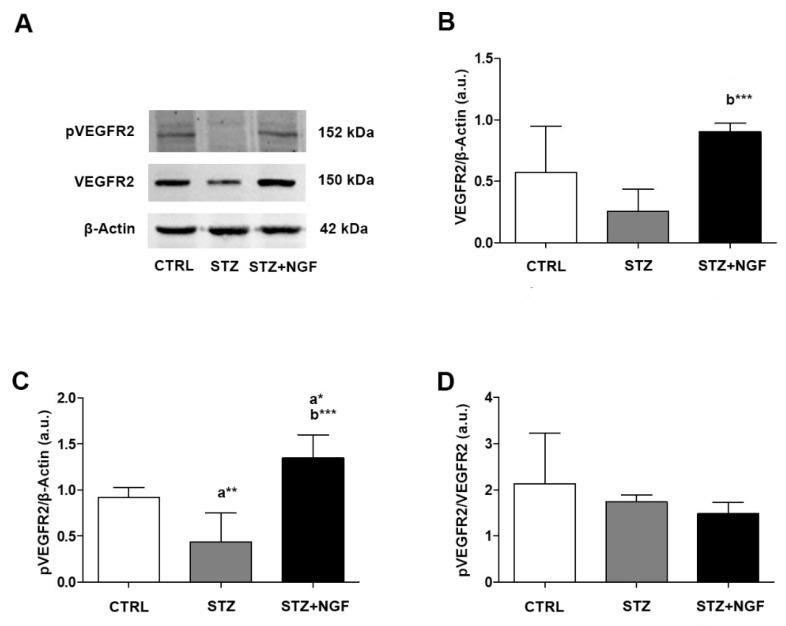
(**A**–**D**). VEGFR2 and its phosphorylated status (pVEGFR2) levels in diabetic rat retina, before and after NGF ocular treatment. VEGFR2 and pVEGFR2 levels altered in STZ and STZ+NGF groups, as showed by Western Blot (**A**) and densitometric analyses (**B**–**D**). β-Actin expression was used to normalize sample variability, a.u. indicates arbitrary units. The results are expressed as mean ±SD. Statistical significance is indicated with * *p* < 0.05, ** *p* < 0.01, *** *p* < 0.001. “a” vs. CTRL group; “b” vs. STZ group. CTRL, healthy rats; STZ, streptozotocin; STZ+NGF, streptozotocin+nerve growth factor eye drops.

**Figure 4 cells-11-03246-f004:**
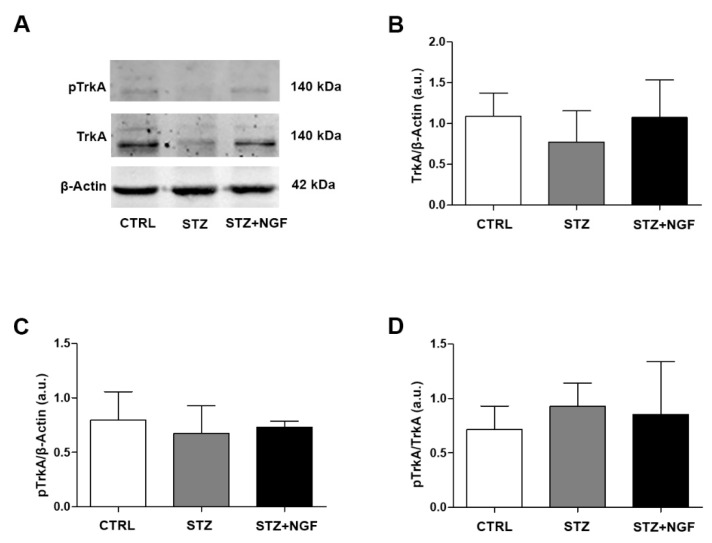
(**A**–**D**). Levels of TrkA and its phosphorylated status (pTrkA) in diabetic rat retina with or without NGF treatment. TrkA and pTrkA levels are shown in the experimental groups by Western Blot (**A**) and densitometric analyses (**B**–**D**). β-Actin expression was used to normalize sample variability, a.u. indicates arbitrary units. The results are expressed as mean ± SD. CTRL, healthy rats; STZ, streptozotocin; STZ+NGF, streptozotocin+nerve growth factor eye drops.

**Figure 5 cells-11-03246-f005:**
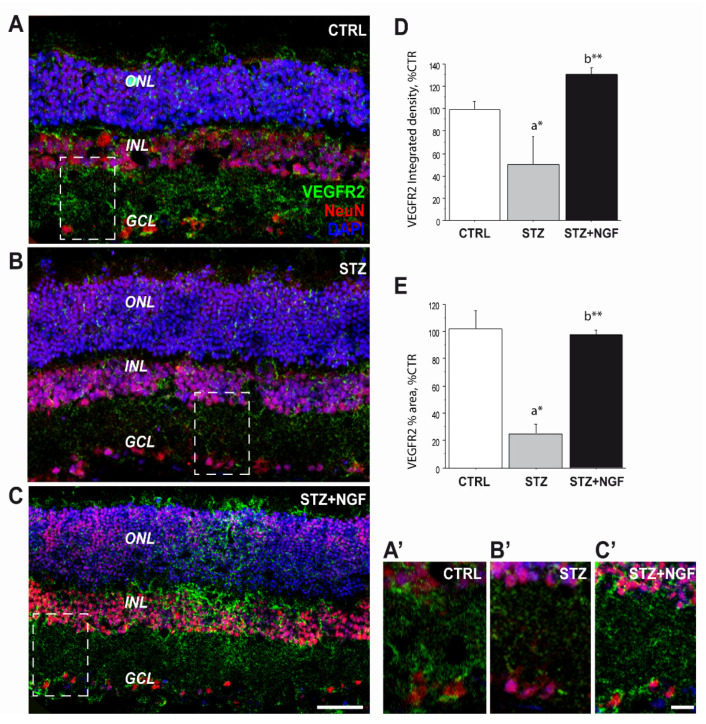
(**A**–**E**). VEGFR2 expression was severely affected upon STZ injection and recovered after ed-NGF treatment. (**A**–**C**) Representative VEGFR2 (green)/NeuN (red) double immunofluorescent staining of the retina from healthy and diabetic rat retina, with or without topical NGF administration. (**A’**–**C’**) The high magnification of the VEGFR2/NeuN staining in the islets showing reduced VEGFR2 expression in the GCL layer of STZ retina, and its recovery after ed-NGF. (**E**–**D**). Quantification of the VEGFR2 immunofluorescence in the retina. The total integrated density (**D**) and percentage of area covered by VEGFR2 staining (**E**) in the three retinal layers. The results are expressed as mean ± SEM. Scale bar: 50 µm; islet: 30 µm. Statistical significance is indicated with * *p* < 0.05, ** *p* < 0.01. “a” vs. CTRL group; “b” vs. STZ group. CTRL, healthy rats; STZ, streptozotocin; STZ+NGF, streptozotocin+nerve growth factor eye drops.

**Figure 6 cells-11-03246-f006:**
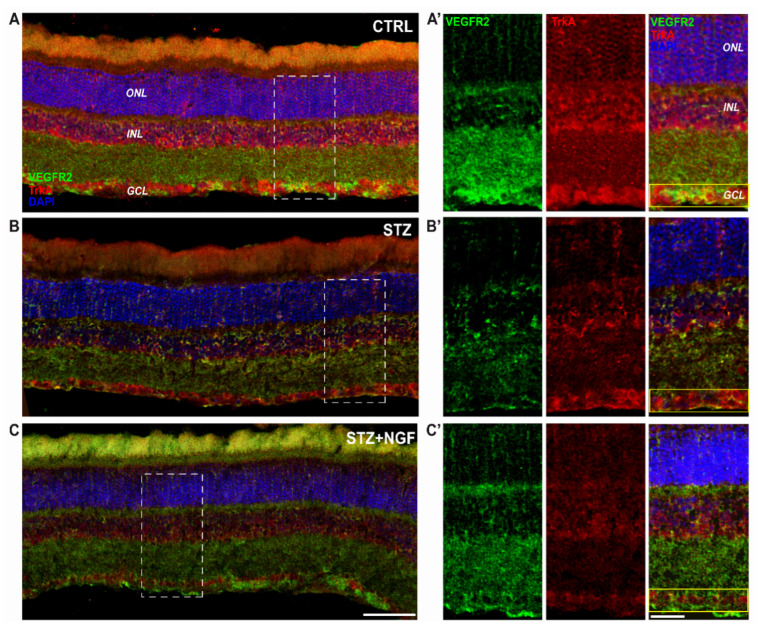
(**A**–**C**). Distribution of TrkA and VEGFR2 in the retinal layers of healthy and diabetic rat retina, with or without topical NGF administration. (**A**–**C**) The immunostaining of rat retinal sections showed the different localization of TrkA (red) and VEGFR2 (green), distributed through the retinal layers: ONL, INL, and GCL. Cell nuclei were labeled by DAPI staining (blue). (**A’**–**C’**) The high magnification reported the separated channels of TrkA/VEGFR2 staining in the retinal layers, focusing on the GCL. Islets corresponding to the scale bar: 100 µm, islet: 30 µm. CTRL, healthy rats; STZ, streptozotocin; STZ+NGF, streptozotocin+nerve growth factor eye drops.

**Figure 7 cells-11-03246-f007:**
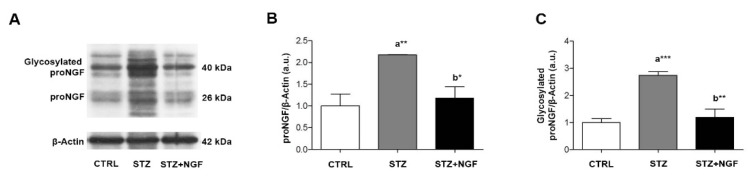
(**A**–**C**). Glycosylated and non-glycosylated proNGF levels in diabetic retina before and after ocular NGF treatment. Altered proNGF levels are shown by Western Blot (**A**) and densitometric analyses (**B**,**C**, respectively). β-Actin expression was used to normalize sample variability, a.u. indicates arbitrary units. The results are expressed as mean ±SD. Statistical significance is indicated with * *p* < 0.05, ** *p* < 0.01, *** *p* < 0.001. “a” vs. CTRL group; “b” vs. STZ group. CTRL, healthy rats; STZ, streptozotocin; STZ+NGF, streptozotocin+nerve growth factor eye drops.

**Figure 8 cells-11-03246-f008:**
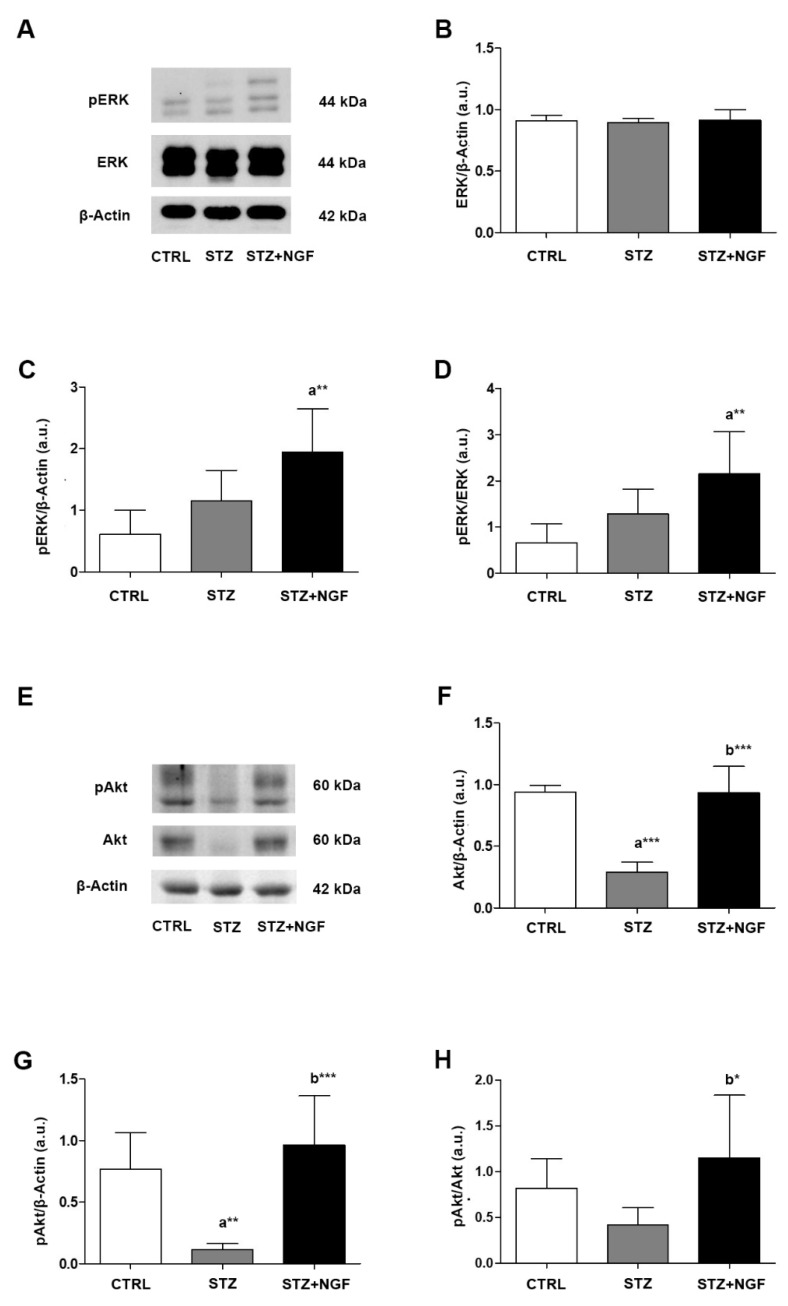
(**A**–**H**). Levels of signaling mediators ERK and Akt in diabetic retinas before and after ocular NGF treatment. ERK and Akt, and their phosphorylation statuses (pERK and pAkt, respectively) levels were affected by diabetes (STZ) and ocular NGF treatment (STZ+NGF). The effects are visible in the representative Western Blots (**A**,**E**) and densitometric analyses (**B**–**D**,**F**–**H**). β-Actin expression was used to normalize sample variability, a.u. indicates arbitrary units. The results are expressed as mean ±SD. * *p* < 0.05, ** *p* < 0.01, *** *p* < 0.001. “a” vs. CTRL group; “b” vs. STZ group. CTRL, healthy rats; STZ, streptozotocin; STZ+NGF, streptozotocin+nerve growth factor eye drops.

## Data Availability

The data presented in this study are available on request from the corresponding author.

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
