# Peer review of "NGF Prevents Loss of TrkA/VEGFR2 Cells, and VEGF Isoform Dysregulation in the Retina of Adult Diabetic Rats"

_cells, 2022, doi:10.3390/cells11203246_

Round 1

Reviewer 1 Report

The purpose of this study was to investigate the effect of NGF eye drops treatment on diabetic rats (STZ injection). Male SD rats were induced to become diabetic by STZ injection, and then treated by NGF eye drops for 2 weeks, starting at 6 weeks post diabetic induction.

Retinas were analyzed by histology, immunohistochemistry, and Western blots.

The data show that the 2-week NGF eye drop treatment reversed vascular changes;, appeared to prevent cell loss, reversed diabetes-induced changes in the expression of VEGFA isoforms, Rock1, and VEGFR2. The expression of VEGFR2 in the ganglion cell layer in reduced in diabetes; this loss is reversed with NGF treatment. The diabetes-induced increase in pro-NGF levels were reversed by NGF treatment. The authors also investigated intracellular signaling pathways (ERK, pERK, Akt, pAkt). They show a significant increase in pERK (but not ERK) in the STZ+NGF group. Akt and pAkt is depressed in diabetes; but returned to normal with NGF treatment.

General comments: This is a well-written paper.  The results are interesting, although they only show a beneficial effect of NGF eyedrops in early stage of diabetes (6 weeks after onset). The authors mention this in the discussion. The reviewer has only few comments.

Specific comments:

1) Were the eyes of each animals divided up for biochemical or histology (e.g., left eye vs right eye), or were separate groups of animals used?

2) What was the reason for not counting cell layers to quantitatively analyze cell loss? It should be relatively easy to count corresponding sections in ImageJ if they are available, or alternatively quantify staining for retinal ganglion cells..

3) It would have been interesting to add TUNEL staining.

4) Materials and Methods, p.5, 1st line: the word “male” is repeated twice in this sentence.

5) Only male rats were used in this study. Can the authors add a justification for only using males?

Author Response

1)Were the eyes of each animals divided up for biochemical or histology (e.g., left eye vs right eye), or were separate groups of animals used?

The
Authors apologise for the lack of this information. Actually since the NGF treatment was performed on both eyes, as reported in Materials and Methods section, and the STZ injection induces a bilateral retina damage, the eyes from each rats were randomly assigned one to the biochemical and one to the histological
evaluation. This information has been incorporated in the
revised version of the manuscript.

2) What was the reason for not counting cell layers to quantitatively analyze cell loss? It should be relatively
easy to count corresponding sections in ImageJ if they are available, or alternatively quantify staining for retinal ganglion cells..

3) It
would have been interesting to add TUNEL staining.

The
Authors appreciate the reviewer’ suggestions, but since the RGC loss in STZ retina have been largely reported (Ken and Barber, 2008; Yang et al., 2012), and NGF eye drops has been previously demonstrated to counteract RGC loss in diabetic rats, as well as in ONC model of retina degeneration (Rosso et al., 2021), we did not focus our study on the identification of the ongoing apoptotic cells (TUNEL) and/or the quantification of RGC loss.

Based on the data presented, and the
literature (Penn JS et al., 2008; Le YZ et al., 2017), which showed that in the retina total VEGFR2 is expressed by different cell type, including Müller cells, the possibility that other cells might be NGF eye drops target, and therefore contribute to the NGF-induced recovery, is currently investigated in our laboratory.

4)
Materials and Methods, p.5, 1st line: the word “male” is repeated twice in this sentence.

Th
e Authors thank for the comment; the repeated term has been corrected.

5) Only male rats were used in this study. Can the authors add a justification for only using males?

Neurotrophins and their receptors can variate across the estrous cycle in both the central nervous system and peripheral tissues (Kaur G et al., 2007; Tirassa P et al., 2015). Thus, adult female animals included in the experiments have to be in the same phases of the cycles in order to avoid overlapping effects due to changes of hormone levels. The rat entire cycle is short (4-5 day) and its staging by vaginal smear is very laborious, increasing the time consuming of the experiment. Therefore, if the evaluation of gender difference in the response to treatments, as in our experimental condition, is not a specific question to be addressed, we usually privilege investigation on adult male animals. The justification for the use of male rats in our study is
incorporated at the beginning of the first paragraph of Materials and Methods.

Reviewer 2 Report

The MS by Fico et al. deals with the mechanisms involved in the effect of eye drops of NGF on STZ diabetic rat eyes.

The MS is a well designed study with appropriate chosen protein targets to be studied. The results are sound and contribute to confirm the role of neurotrophins in supporting viability of neural retinal cells. The conclusions are supported by the findings reported.

English language requires revision.

Author Response

The Authors thank the Reviewer for the comments and suggestions.